# Efficacy of Polymer-Based Nanomedicine for the Treatment of Brain Cancer

**DOI:** 10.3390/pharmaceutics14051048

**Published:** 2022-05-13

**Authors:** Tobeka Naki, Blessing A. Aderibigbe

**Affiliations:** Department of Chemistry, University of Fort Hare, Alice Campus, Eastern Cape 5700, South Africa; baderibigbe@ufh.ac.za

**Keywords:** nanomedicine, drug delivery, brain cancer, tumor, blood–brain barrier, anticancer drugs

## Abstract

Malignant brain tumor is a life-threatening disease with a low survival rate. The therapies available for the treatment of brain tumor is limited by poor uptake via the blood–brain barrier. The challenges with the chemotherapeutics used for the treatment of brain tumors are poor distribution, drug toxicity, and their inability to pass via the blood–brain barrier, etc. Several researchers have investigated the potential of nanomedicines for the treatment of brain cancer. Nanomedicines are designed with nanosize particle sizes with a large surface area and are loaded with bioactive agents via encapsulation, immersion, conjugation, etc. Some nanomedicines have been approved for clinical use. The most crucial part of nanomedicine is that they promote drug delivery across the blood–brain barrier, display excellent specificity, reduce drug toxicity, enhance drug bioavailability, and promote targeted drug release mechanisms. The aforementioned features make them promising therapeutics for brain targeting. This review reports the in vitro and in vivo results of nanomedicines designed for the treatment of brain cancers.

## 1. Introduction

Cancer is a class of diseases characterized by uncontrolled cell growth and, in some cases, spreads to the surrounding tissues [1]. The uncontrolled growth of cells in the brain is known as brain cancer. There are two grades of brain cancer, namely primary brain cancer that starts in the brain and secondary brain cancer that originates elsewhere in the body and spreads through the bloodstream [2]. There are two major forms of brain cancer: benign brain cancer (which is slow-growing, such as meningiomas, craniopharyngiomas, pituitary tumors, neuromas etc.) and malignant brain tumors (which spread to other parts of the brain or spinal cord, e.g., astrocytomas, glioblastomas, oligodendrogliomas, etc.) [3]. There are also different grades of brain cancer based on the appearance of the cells such as grades I–IV. Grade I cells resemble normal brain cells and spread slowly; grade II cells are abnormal but grow slowly; grade III cells are abnormal and spread to the nearby tissues; and grade IV cells are abnormal and grow at a rapid rate [4,5].

Brain cancer is the cause of a large number of deaths. In 2012, there were 116,605 females and 139,608 males treated with primary malignant brain and other CNS tumors globally [6]. There were about 18,078,957 cases of cancers worldwide according to the 2018 Cancer Registry with approximately 296,851 cases associated with brain cancers. The highest number of cases of brain cancer was reported in Asia (156,217) and the lowest number was reported in Oceania (2438 cases). In 2018, there were about 9,555,027 cases of death around the world, of which 241,037 deaths were related to brain cancers [7]. In terms of global cancer statistics, GLOBOCAN 2020 confirmed 308,102 cases of brain cancer with 251,329 deaths in 2020, revealing that the brain cancer mortality rate is increasing yearly [8]. The 10- and 5-year survival rates for malignant brain and other CNS tumors have been confirmed to be 34.9% and 29%, respectively. The survival rates vary depending on the tumor’s histological classification [9].

The major challenge in the development of effective therapies for the treatment of brain tumor is the heterogeneity of the brain tumors and the BBB [10]. To overcome these challenges, researchers have developed nanomedicine with innovations translated to clinical applications. Nanomedicine offers the possibility of developing sophisticated targeting therapeutics with multi-functionality [11]. Nanomedicines have been reported to accumulate selectively in the tumors due to the enhanced permeability and retention (EPR) effects, which takes full advantage of cancer tissue permeable vasculature and reduced lymphatic drainage. However, only a small amount of the total administered drug accumulates in the tumor site via this pathway. Nanomedicine designed with ligands for targeted drug uptake for enhanced drug accumulation is an interesting approach [12]. The advancement in the development of nanomedicines for cancer therapy over the last ten years has yielded promising results with the potential to be translated to clinical application. Hydrophobic and hydrophilic anticancer compounds have been incorporated into nanocarriers resulting in sustained and controlled drug release profiles suitable to overcome drug resistance and toxicity. Over the last ten years, polymeric nanomedicines have emerged as a good platform for the treatment of malignant tumors. Polymeric drug carriers, such as polymersomes, nanogels, polymeric micelles, etc. can improve drug stability, extend drug circulation time, minimize side effects, enhance tumor accumulation, and regulate the release of bioactive agents [13].

## 2. Brain Targeting

### 2.1. Transport Mechanisms

Drug molecules travel from the nasal cavity to the brain parenchyma via the trigeminal or olfactory nerves. The molecules are also transported to the roots of nerves in the cerebrum and pons after spreading across the brain. The mechanism is via intracellular and extracellular pathways [14]. In the intracellular pathway, the uptake of the drug molecule is via the olfactory neuron and the drug release is via exocytosis. In the extracellular pathway, the drug uptake to the lamina propria is via the nasal epithelium followed by transportation through the neuronal axon by processes known as bulk flow. It is important to note that the axon leads into the CNS, where the drug is released and transported further through a fluid movement [14].

#### 2.1.1. Pathways of Nasal Transportation to the Brain

The first set of structures for the respiratory system is formed by the interior part of the nose or nasal cavity, i.e., the entry point for inhaled air [15]. The nasal cavity is separated into two compartments by the nasal septum. The mucosal layer covers the cavity and protects against pathogens that are infectious and allergic [15]. The nasal cavity is made up of the respiratory section (a passage for airflow), the nasal vestibule (a wide area of the nostril), and the olfactory section (composed of olfactory receptors) [16]. The respiratory section is also involved in drug uptake. It promotes drug uptake because it is highly vascularized, making it useful for nose-to-brain transportation of drugs [17]. The vestibule is not vascularized and also not permeable to drug uptake [17]. The mucosa on the olfactory is composed of nerves connected to the olfactory nerve [17]. The cell type in the vestibule is known as the squamous epithelial cells, composed of a few ciliated cells with small surface areas that limit drug absorption. The respiratory section has a large surface area and contains cell types such as ciliated, non-ciliated columnar, goblet, and basal cells [18]. The goblet cells produce mucin that creates the mucus layer. The mucus traps drug molecules and delivers them to the throat for uptake into the GI tract. Drug uptake via the mucus layer can reach the surface of the epithelium. However, the viscosity of the mucus influences the drug clearance rate. The higher viscosity of the mucus results in a low clearance rate and high uptake of drugs. A combination of the highly vascularized and the large surface area of the respiratory section makes it a significant site for drug absorption to the systemic circulation [18]. The therapeutic agents may be absorbed and transported via a neuronal pathway, such as the trigeminal nerves or olfactory or vasculature, and through pathways including the lymphatic system or CSF [19]. One or more pathways may be used to transfer drugs from the nasal cavity to the brain.

#### 2.1.2. Olfactory Neuronal Pathways

The olfactory neuronal pathway is considered to be one of the most effective intranasal drug absorption pathways in the brain. The drug molecules cross the olfactory epithelium [20]. The drug uptake across the cell membrane is controlled in three strategies: passive diffusion, neuronal endocytosis, and paracellular motion. Lipophilic drug uptake occurs via passive diffusion. These pathways of drug uptake are impacted by the molecular weight of a drug molecule and lipophilicity [21,22].

#### 2.1.3. Trigeminal Nerve Pathways

Intracellular transport via endocytosis or axonal transport is used to move drugs from the nose via the trigeminal nerve pathway [23]. The largest cranial nerve is the trigeminal nerve. The ophthalmic, maxillary, and mandibular are all part of it. In nose-to-brain drug transmission, the ophthalmic and maxillary branches are important. The dorsal section of the nasal mucosa and the anterior nostril is innervated by the trigeminal nerve’s ophthalmic branch. The nasal mucosa turbinates are innervated by the maxillary branch [23,24]. When a drug molecule disperses via the nasal mucosa, it goes to the branches of trigeminal nerves in the respiratory and olfactory regions, as well as to the axonal pathway through the brain stem. Drug transmission from the nasal cavity to the forebrain is also aided by a portion of the trigeminal nerve that crosses through the cribriform plate [25]. Paracellular, transcellular, receptor-mediated transport, carrier-mediated transport, and transcytosis have all been reported to be involved in drug transportation via the mucosa [26]. Drug molecules are conveyed between cells via the paracellular route, whereas drugs are delivered across cells via the transcellular route. Endocytosis or carrier-mediated transport are two options for transcellular transportation. Adsorptive transcytosis, which involves an interaction between the cell surface and the ligand in the bloodstream, is used to transport macromolecules via the transcellular pathway. The interaction between the negatively charged membrane and the positively charged macromolecules and proteins is electrostatic. Transcytosis is a major transportation approach of nanoparticles and chemicals to the CNS [22,25].

#### 2.1.4. Cerebrospinal Fluid (CSF) Pathways

The cerebrospinal fluid pathways run from the CSF of the brain’s subarachnoid space to the nasal lymphatic system via the olfactory nerves located in the perineurial space [22]. However, there is no detailed research report on drug delivery in the CSF lymphatic passageway from the nose to the brain. There is a need for more investigations. In a study by Johnston et al., radioactively labelled tracer filled into the CSF was conveyed through the olfactory nerve-associated passages to the nasal lymphatic and cervical lymph node system [22]. This pathway is a potential route for the transportation of drugs to the brain. Factors such as the drug molecule’s molecular weight, the degree of ionization, and the lipophilic nature of the drug influence the drug uptake in this pathway [27]. The lipophilic drugs are distributed more effectively [22]. This pathway is a potential route for the transportation of drugs to the brain.

## 3. Drug Delivery to the Brain

There are reports on the design of therapeutics for brain targeting. A 29-amino-acid peptide (RVG29) obtained from rabies virus glycoprotein was used as the targeting ligand in the brain-targeted drug delivery system. In vitro studies showed that the drug delivery system, RVG29-lip, exhibited remarkably greater uptake in dopaminergic cells and murine brain endothelial cells as well as via the BBB. In a Parkinson’s disease (PD) mouse model, the therapeutic did not induce systemic toxicity after intravenous administration [28]. A variety of nanoparticles capable of transporting drug payloads to specific organs or tissues have been reported by several researchers. Their capability to transport drugs to the diseased tissues/organs makes them appropriate for targeted drug delivery. However, targeted drug delivery requires the conjugation of a receptor-specific ligand to the surface of the nanoparticles. Nanoparticles are characterized by a large surface area suitable for interaction with the cells in non-specific surface interactions [28].

Complex drug delivery systems are generally costly, adding to the difficulty of maintaining a batch-to-batch consistency. Nanocarriers can be enhanced to promote passive and active targeting mechanisms for enhanced uptake into the tumor tissue. The efficacy of bortezomib in the treatment of glioblastoma was reported by drug administration into the brain lesion via catheter when compared to intravenous systemic delivery [29]. A double emulsion solvent evaporation method was used to develop selegiline-loaded poly (lactic-co-glycolic acid) nanoparticles loaded in a transdermal film. The in vivo study revealed good brain targeting capability of the formulation for over 72 h [30]. Intranasal delivery has been reported to bypass the BBB and transport the drugs into the CNS at a faster rate. Mucoadhesive drug delivery techniques increase the drug’s time at the site of application. They also improve the formulation’s interaction with the absorption surface, which enhances the drug’s therapeutic results [31]. In vivo, zaleplon encapsulated into an in situ nanoemulsion gel with droplet sizes ranging from 35 to 73 nm increased drug bioavailability by eightfold. With high systemic bioavailability, nasal tissue penetration was also improved [32]. In situ gelling formulations for intranasal administration prepared from deacetylated gellan gum and loaded with curcumin showed good bioavailability. The uptake of curcumin and its distribution in the brain was enhanced when compared to the intravenous injection. The brain targeting index of the intranasal gel formulation was 0.39, which was higher than the intravenous injection, i.e., 0.06 [33]. Other formulations, such as functionalized risperidone liposomes for nasal delivery of risperidone to the brain, have been reported to be a promising therapeutic for the management of schizophrenia. Liposomal formulations promoted better brain uptake in vivo. Furthermore, risperidone uptake into the brain was higher from the PEGylated liposomes, (LP-16) than in the plasma. LP-16 displayed a high brain targeting efficiency index, suggesting that the drug uptake to the brain is selective. Some researchers have effectively formulated surface-modified risperidone liposomes for nasal delivery with good brain targeting delivery [34]. Magnetic nanoparticles have been designed for brain targeting. It was prepared by single emulsion solvent evaporation of polymers with oleic acid-coated magnetic nanoparticles and Rh123. The capability of magnetically targeted nanoparticles to transmit substances to the brain was established. The uptake of the nanoparticles revealed their capability to overcome the P-gp efflux system [35].

Bortezomib (BZM) has been studied for its efficacy via intranasal delivery to the brain by a combination with NE100 (enriched perillyl alcohol). The intranasal delivery of BZM in combination with NE100 increased the survival rate of the tumor-bearing animals compared to those who received administered BZM alone. Furthermore, intranasal delivery of NE100 in combination with BZM promoted the high concentration of the drug in the brain [28].

Following TDDS application, the drug targeting efficacy and ability of selegiline hydrochloride-loaded PLGA nanoparticles (SGN-NPs) were computed as per cent. Drug targeting effectiveness (DTE) and drug targeting potential were also studied. According to the results, nanoparticles have a higher DTE (136.87 ± 3.84%) and drug targeting potential (26.93 ± 4.29%) than the plain drug, implying that nanoparticles have a stronger brain targeting performance. This is attributed to the loading of SGN into PLGA nanoparticles that effectively cross the BBB to deliver the drug to brain tissue [29]. The use of a facial intradermal injection to circumvent the BBB through the trigeminal nerve has been reported to be an innovative approach to bypassing the BBB. Intradermal injection of Evans blue (EB) into the rat mystacial pad resulted in increased drug concentration of Evans blue in the brain sub-areas when compared to administration via intravenous (i.v.) and intranasal (i.n.) injections. The intradermal injection has also improved brain drug targeting effectiveness, brain direct transfer percentage, and brain bioavailability of EB, even though intranasal inoculation only slightly changed them. The facial intradermal injection, which bypasses the BBB through the trigeminal substructures, is a potential method for brain-targeting delivery [36]. Polyethene glycol, cholesterol, and 1,2-distearyl-sn-glycerol-3-phosphocholine were used to generate liposomes by Al Asmari et al. (DSPC). Liposomal formulations of donepezil administered through the intranasal route enhanced the bioavailability of donepezil in the brain and plasma [37]. In another analysis, a ligand (the RVG29 peptide) that could bind to acetylcholine receptors was coupled to polyethylene glycol-modified poly-(d,l-lactide-co-glycolide) to create a targeted conveyor; nanoprecipitation was used to make the targeted docetaxel nanoparticles (DTX-NPs). The NPs had smooth surfaces and were about 110 nm in diameter. The number of receptors on the surface of glioma cells was 2.04-fold greater than that of non-malignant cells and it facilitated the uptake of RVG29-improved NPs at the targeting site. In an in vitro cellular uptake analysis, the NPs exhibited targeting uptake in the glioma cells compared to the non-targeting NPs. Targeted NPs displayed greater BBB infiltration in an in vitro model. RVG29-PEG-PLGA-NPs were found to be taken up in the intracranial glioma tissue in vivo studies. Furthermore, findings suggested that RVG29-modified NPs are useful for the treatment of glioma [36].

Neurotropic viruses, such as rabies, can penetrate the BBB and infect brain cells. Peptidyl-targeting vectors based on rabies virus glycoprotein (RVG) that had previously been exhibited to bind to the nicotinic acetylcholine receptor in the neurons were developed. Intravenous administration of the formulation in mice resulted in RVG29, i.e., 29-mer peptide, and exhibited selective brain uptake, resulting in a three-fold increase in RVG29 uptake in the brain compared to the mock peptide. In addition, immunocompromised mice were injected with the Japanese encephalitis virus and the therapeutic efficacy of RVG29-mediated siRNA delivery was discovered. The antiviral RVG29 siRNA complex was administered to the infected mice intravenously once every day for 3 days post-infection. At 30 days after infection, medicated mice had an 80% survival rate, while all control mice died within ten days [37,38,39].

The capability of biodegradable poly (N-butyl cyanoacrylate) (PBCA) nanoparticles to cross the BBB has been widely reported. As opposed to the free drug, PBCA coated with polysorbate 80 raised the concentration of rivastigmine in the brain by 3.8 times. Liposomes loaded with both transferrin (T) and folate (F) were conjugated to create dual-targeting doxorubicin (DOX) liposomes (Tf). The findings showed that they were able to penetrate the BBB and target brain tumors. Yang et al. used a transfection reagent to separate exosomes from brain EC culture media and loaded them with vascular endothelial growth factor siRNA. In xenotransplanted zebrafish carrying brain tumors, the exosomes enhanced siRNA passage via the BBB [40].

Jinbing et al. used magnetic nanoparticles, i.e., PLGA, L-a-phosphatidylethanolamine (DOPE), and 1,2-distearoyl-sn-glycero-3-phosphoethanolamine-N-amino (polyethylene glycol) (DSPE-PEG-NH2), to design magnetic poly (D, L-lactide-co-glycolide) (PLGA)–lipid (MPL) nanoparticles (NPs). TAT-MPLs were designed by conjugating the trans-activating transcriptor (TAT) peptide into the surface of MPLs, prompting them to target the brain using magnetic guidance and TAT penetration. The great fluorescence intensity in the cytoplasm and cell nucleus suggested that TAT-MPLs loaded into QDs were effectively delivered into the BECs. The coupling of TAT on MPLs remarkably enhanced the cellular absorption of administered drugs in bEnd.3 cells by increasing cell membrane infiltration, suggesting the successful BBB crossing of the drug delivery systems [41]. A polyanhydride polymer wafer comprising bis-chloroethyl nitrosourea (BCNU) (carmustine) was used to treat recurrent high-grade gliomas. A 2-month-long drug release was reported. Despite the increased risk of trauma and the ineffectiveness of the distribution system, the formulation was reported to be a promising system [42]. Protein complex nanoparticles made of poly (ethylene glycol)-poly (lactic acid) block copolymer (PEG-PLA) were unable to pass the safe BBB. On the other hand, the complex nanoparticles delivered brain-derived neurotrophic factor (BDNF) to the brain and improved efficiency in a mid-cerebral artery occlusion mouse replica for strokes. Furthermore, the PEG-PLA BDNF complex reached the brain through a stroke-induced disruption of the BBB [40].

### 3.1. Nanodelivery Platforms Developed for Drug Delivery

#### 3.1.1. Passive Targeting

Hyper-vascularizing, leaky, and scarce lymphatic drainage systems are properties that promote passive targeting, thereby enabling the drug uptake into the intratumoral space while protecting the healthy brain tissue (Figure 1). Passive targeting causes rapid and damaged angiogenesis, and blood vessels in tumors can possess a dripping endothelium that fails to perform its usual barrier function, enabling macromolecules as small as 400 nm to enter. The nanoparticles passively push through into tumor tissue via dripping vasculature, collecting in the tumor bed due to defective lymphatic drainage, releasing therapeutic payloads into the area of tumor cells during the intravenous administration period. The enhanced permeability and retention (EPR) effects are then given to this process. Furthermore, the EPR effect is used in most nanomedicines that are presently in agreement for clinical applications in therapy for solid tumors. According to the literature, the optimum size of NPs should be below 100 nm and the existence of a hydrophilic surface to circumvent clearance by macrophages inside the reticuloendothelial system should be considered. When the nanoparticles measure was below 100 nm, good results were observed. However, passive targeting strategies have some restrictions since the EPR effect is completely dependent on drug diffusion, which is hard to monitor, making it unlikely for some drugs to disperse efficiently. Due to a thick brain matrix, which obstructs diffusion and increases interstitial fluid pressure, the EPR effect on brain tumors is unlikely to be well organized [43,44,45]. Generally, nanoscale drug delivery systems accumulate in the tumor via the porous brain tumor–blood barrier (BTBB) through EPR effects and are retained due to reduced lymphatic drainage. The EPR effect applies to nanosized carriers in the range of 10–200 nm [36]. Many others have found that this passive targeting alone cannot overcome the BBB or does so to a degree that is insufficient for effective treatment, showing that only smaller particles (approximately 20 nm) or smaller (<12 nm) [37] can cross the BTBB [38,39]. Although studies have found that small nanoparticles and drugs can take advantage of the leaky tumor vasculature [42], various methods to further improve penetration of the BTBB should be explored.

#### 3.1.2. Active Targeting

Active targeting involves the use of a mixture of nanocarriers and ligands that bind with various cell surface receptors. Active targeting of the BBB is a non-invasive and promising approach for drug delivery to brain tumors. To obtain drug delivery, active targeting of the tumor sites enable the use of an intrinsic cell characteristic (Figure 2). A homing system, such as an antibody or ligand, is also used. Many novels and smart devices have been used. Drug uptake into tumors via antigens or receptors without impacting normal tissues has been reported. As a result of external stimuli, such as heat, ultrasound, light, and a magnetic field, environmentally sensitive macromolecular drug carriers can release cargo drugs in the targeted tumor tissues. In comparison to other active targeting methods, this triggered drug release has many advantages. It also allows for precise control over drug release timing and position. Active targeting has been effective in producing a series of antibody–drug conjugates (ADC) due to their selectivity and binding affinity. However, the ADC approach is limited by some challenges that must be resolved to take a greater step forward [46,47,48].

Although external stimuli can release cargo drugs in the targeted tumor tissues, there are also limitations associated with their use, such as susceptibility to off-target delivery in pH-sensitive systems, limited penetration of ultrasound to deep tissues by acoustic-based drug delivery systems, prolonged exposure time resulting in damage to the surrounding healthy tissue in light-sensitive systems, and complications which require a sensitive synthesis approach [49,50,51].

#### 3.1.3. Mechanisms of Efflux in Drug Transport to the Brain

BBB absorption and efflux processes are useful for drug targeting to the brain and the desired CNS pharmacological impact. Drug penetration through the BBB is reduced to avoid exposing the CNS to the side effects. Most in vivo methods for drug absorption into the brain would immediately include incorporating any task of CNS efflux. The CNS has many efflux pathways that impact drug concentrations in the brain. Other mechanisms are active and some are passive. Active efflux from the CNS through particular transporters can frequently minimize measured drug absorption at the BBB to levels below those anticipated by the drug’s physicochemical properties, such as lipid solubility. The amount of free drugs obtainable to interact with drug receptor sites in the brain extracellular fluid is influenced by the function of these efflux mechanisms.

The P-glycoprotein (Pgp), the multidrug transporter, the multi-specific organic anion transporter (MOAT), and the multidrug resistance protein (MRP) are all members of the ABC cassette of transport proteins, which are currently receiving a lot of attention. In humans, Pgp is the outcome of the MDR gene, which recognizes a broad variety of lipid-soluble substrates and actively effluxes them from cells, indicating the gene product. MOAT in the choroid plexus has substrate preferences that are similar to MRP. Co-administration of a Pgp inhibitor with certain Pgp substrates can not only improve oral absorption but also BBB porosity. In mice, co-administration of the Pgp blocker valspodar not only increased the brain levels of paclitaxel but also significantly improved its therapeutic consequence on tumor volume [52].

## 4. Mode of Transportation across the Brain

### 4.1. Suggested Mechanisms of Transport through the Blood–Brain Barrier

BBB is mainly built up of a polarized layer of vascular endothelial cells with astrocytes and tight junctions. The different transcellular transport actions can be differentiated into the following processes. (i) Diffusion that is operated by a concentration gradient, including tiny hydrophobic molecules. Some therapeutics use this mechanism as their key entry point into the brain. (ii) Paracellular transport is restricted to tiny water-soluble molecules. (c) Proton pumps efflux transporters. (d) Adsorptive transcytosis presumably mediated transport which is used for positively charged cargo in a non-specific way. (e) Receptor-mediated transcytosis is a peptidic signalling pathway linked to regulatory molecules, such as leptin, insulin, interleukins, and nutrients. (f) Carrier-mediated transport takes place, for example, in amino acids, nucleosides, glucose, and curative, (including azidothymidine vinca alkaloids, etc.). Focusing on receptors that internalize ligands through vesicular transport is considered to be a good choice, such as the intracellular transport containers that can easily accommodate nanoparticles as small as 100–200 nm. This form of transcellular transfer is very normal in polarized cells. In polarized cells, transcytosis is the vectorial transport of cargo in the middle of the apical and basolateral surfaces. Three stages that show different transcellular transport are: (a) intracellular vesicular trafficking towards the opposing surface at which exocytosis of the vesicular contentments occurs; (b) endocytosis of the nanoparticle/cargo at the plasma membrane; and (c) transcytosis, which can depend on receptor-mediated and absorptive charge-dependent endocytic internalization events. For serum albumin, the absorptive manner has been represented. Deltorphin, enkephalin transferrin (iron), insulin, and LDL may all be implicated in receptor-mediated transcytosis in the brain endothelium. This also addresses the possibility of selective targeting of nanoparticles in the brain using a transcytotic passageway, preferably an endocytosis-mediated degradation passageway [53,54,55].

#### 4.1.1. Paracellular Transport

Distinct junctions, such as zonula adherens, close junctions, and macular adherens, connect cells inside the nasal epithelium. Large molecule drugs cannot move through these junctions. However, neuronal and basal cells become permeable because of constant turnover. The opening of these junctions helps paracellular transport. Various DDSs are rapidly transmitted via nasal mucosa to the brain by opening these junctions, according to some reports [56].

#### 4.1.2. Transcellular Diffusion

Davison demonstrated in 1955 that the separation of substances from the blood into the CSF and brain tissue was dictated by lipid solubility and size. Tiny lipophilic molecules can also pass through the BBB via passive transmembrane diffusion, which is now well understood. Most drugs that can reach the CNS do so through passive transmembrane diffusion, which is not a saturated mechanism. The partitioning of lipid-soluble compounds into aqueous and nonpolar media, such as octanol and water, defines their ability to cross the BBB. Furthermore, substances that spread passively through the BBB must move through the cytosol, luminal membrane, and abluminal membrane before reaching the CNS. The lipophilicity of a drug, on the other hand, favors diffusion through the BBB over sequestration within the cell membrane. When the analyzer agent is an efflux transporter substrate, the transcellular diffusion of substances through the BBB is also influenced (Figure 3) [57,58].

## 5. The Different Nanomedicines Designed for Brain Cancer Therapy

### 5.1. Polymer-Based Nanoparticles

Polymer-based nanoparticles (NPs) are nanomedicines with particulate dispersions or solid colloidal particles with a diameter ranging between 1 and 1000 nm [59]. Therapeutic molecules are adsorbed, encapsulated, or conjugated in the polymer matrix of polymeric nanoparticles (Figure 4). Polymeric NPs have a long half-life in the systemic circulation with the improved release of the agents from the NPs. Furthermore, polymeric molecules comprise different dissolvable profiles in different solvents. It is useful to functionalize various delivery and targeting purposes. The polymeric NPs with desired physicochemical properties can be protected from enzymatic degradation, rapid clearance, and hepatic metabolism. The large surface area of polymeric NPs is an appealing characteristic for controlling the release kinetics, drug loading capability, and administration path [60]. Since these brain-penetrating nanoparticles (BPNPs) have the potential for cell targeting and regulated drug release after administration, they have a high potential for treating intracranial tumors [61]. Some of them have been licensed by health regulatory agencies for a clinical trial for the delivery of a variety of therapeutic agents and diagnostic of target organs [60].

As opposed to non-modified nanoparticles, the amount of poly(ethylene glycol) nanoparticles (PEGylated NPs) found in tumors has been enhanced. Other hydrophilic polymers, such as dextrans, heparins, and polyvinylpyrrolidone, can be used to achieve the same result [62]. PEG-PLGA nanoparticles were tested for their ability to deliver anti-glioma drugs by actively targeting the tumor. In contrast to paclitaxel-conjugated nanoparticles and the commercial drug Taxol^®^, the targeted nanosystem demonstrated higher inhibition of tumor and extended the survival time in rats with intracranial C6 glioma [63].

Pharmacokinetic research was used to examine the polymeric nanoparticles (PNP)–sirolimus and the findings demonstrated that PNP–sirolimus circulates in the blood for a long time. In xenograft tumor mice, PNP–sirolimus circulation was maintained and the in vitro killing consequence of free sirolimus against cancer cells, and intravenous administration revealed its powerful in vivo anticancer efficacy. Furthermore, in vitro and in vivo studies using PNP–sirolimus improved the radiotherapeutic potency of sirolimus. According to the clinical application, PNP–sirolimus circulates proved useful for cancer treatment [10]. Nanoprecipitation was used to make surface-improved poly (d,l-lactic-co-glycolic acid) (PLGA) nanoparticles for achieving a therapeutic concentration of paclitaxel (PTX) in brain tumors. In vitro evaluation of the NP cytotoxicity and cellular uptake was performed on C6 glioma cells. Biodistribution and brain penetration were examined in BALB/c mice after intravenous administration. Moreover, the results revealed that nanoprecipitation parameters could be fine-tuned. The PLGA NPs coated with surfactants with a dimension of about 150 nm delivered PTX for more than two weeks. When opposed to non-coated NPs, the surface coatings can maximize cellular uptake performance, with D-tocopherol polyethylene glycol 1000 succinate (TPGS) showing the greatest improvement. When compared to bare Taxol^®^ and NPs, TPGS-PLGA NPs showed a greater accumulation of PTX in the brain tissue (>800 per cent after 96 h). Finally, PLGA-TPGS with PLGA-NPs coating were effective for good transportation of PTX through the BBB, with the benefits of simple formulation, low production costs, and greater encapsulation reliability [64].

For the transportation of rhodamine -123 and loperamide into the brain, PLGA NPs have been conjugated with a glyco-heptapeptide. Glyco-heptapeptide coating imitates the behavior of opioid peptides through absorption-mediated endocytosis. These NPs were allowed to move through the BBB with simplicity [65]. Polymer nanoparticles, such as block copolymer nanoparticles and poly (lactic-co-glycolic acid) nanoparticles comprising of poly (ω-pentadecalactone-co-p-dioxanone) and polyethylene glycol, hybrid nanoparticles using poly (lactic-co-glycolic acid), charged 1,2-dioleoyl-3-trimethylammonium-propane, and 1,2-distearoyl-sn-glycero-3-phosphoethanolamine-N-(carboxy-poly(ethylene glycol)), were applied for the delivery of distinct anticancer drugs, displaying enhanced drug release and targeting effectiveness with a low brain tumor dimension [66].

In the in vitro evaluation of human primary brain cancer cells with hybrid polymer–nucleic acid–gold nanoparticles, the cells internalize the nanoparticles and transported them to the cytoplasm. Furthermore, these hybrid polymer–nucleic acid–gold nanoparticles are a theranostic platform technology for delivering genetic therapy combinations to human cells [67]. In comparison to the clinically applied DTX formulation Taxotere^®^, an intravenous injection of DTX-NPs raised the blood circulation period of DTX by 5.5 times and the AUC0-24 h in the tumor-bearing brain by 5-fold. Ex vivo fluorescence imaging, on the other hand, was applied to analyze the kinetics of NPs in the brain. This finding supported the delivery of DTX into the brain by NPs and indicated that ex vivo fluorescence imaging of NPs is a useful and fast way to assess drug disposition in the brain. This finding supported the delivery of DTX into the brain by NPs and indicated that ex vivo fluorescence imaging of NPs may be a useful and fast way to assess drug disposition in the brain [67,68].

Nanoparticles of poly (beta-amino ester) (PBAE) were developed. Transfected cells experienced controlled apoptosis as a result of their plasmid delivery of a suicide gene treatment to pediatric brain cancer replicas, precisely the herpes simplex virus type I thymidine kinase (HSVtk). In mice inserted with AT/RT (MB (*p* = 0.0001 vs. control and *p* = 0.0083 vs. control), PBAE-HSVtk medicated categories had a higher median overall survival. Their findings show that biodegradable PBAE nanoparticles can be used as secured and efficient nanomedicine for pediatric CNS malignancies [69]. Ahmed et al. prepared terpolymer–lipid hybrid nanoparticles with doxorubicin (DOX–TPLN). In vitro results revealed sevenfold higher effectiveness of DOX–TPLN towards human GBM U87-MG-RED- Fluc cells as compared to free DOX [70].

Combining information from multiple fields such as cell biology, chemistry, and tumor pathophysiology could help enhance the clinical translation of polymer-based nanoparticles. A detailed understanding of how nanoparticle modifications influence biological systems is important to improve the conjugate design with effective therapeutic effects [71].

The polymer-based NPs possess the ability to be loaded with various types of anticancer agents for the treatment of brain cancer. Some of the in vivo studies exhibited that drug-loaded NPs can result in high tumor inhibition and prolonged survival time of animals with brain cancer in vivo. Nanoparticles can also improve pharmacological parameters of the loaded drug, such as extended time of drug circulation in the blood until they reach the targeted brain tumor. The in vitro studies demonstrated that polymeric NPs can increase cellular uptake of anticancer agents by brain cancer cell lines, indicating that they can lead to excellent antitumor activity. More interestingly, these nanomedicines can simply pass through the BBB, which is the most important mechanism in the treatment of brain cancer and other brain-related diseases.

### 5.2. Nanoliposomes

Nanoliposomes are artificial and small vesicles that are spherically shaped [72]. These nanomedicines are mainly formulated from cholesterol and phospholipids, and they display both hydrophobic and hydrophilic properties with excellent biocompatibility. The nature of the nanoliposome bilayers, such as bilayer charge, permeability, and rigidity, is influenced by the components used for their formulation. The particle sizes of nanoliposomes are in the range between 30 nm and several micrometres [73]. These nanomedicines are used as drug delivery systems for hydrophobic and hydrophilic drugs. They possess the ability to be encapsulated with hydrophilic and lipophilic bioactive agents due to their biocompatibility, biodegradability, and non-toxicity. Furthermore, nanoliposomes promote site-specific drug delivery which makes them useful for the delivery of bioactive agents to the brain [73]. Wang et al. demonstrated the impact of liposomes when administered alone. The liposome did not induce any significant necrosis or apoptosis in C6 glioma cells. However, loading quercetin into nanoliposomal formulation resulted in enhanced antitumor activity in C6 glioma cells [74]. Clinical reports on liposomes show that they are safe and effective when encapsulated with drugs by reducing their off-target accumulation and extending circulation time [75]. Many commercial liposome products are manufactured on industrial scales and their synthetic nature makes it easier to maintain their batch-to-batch consistency [44,76].

Liposomes containing transferrin–folate doxorubicin were examined. The findings indicated that the amount of doxorubicin transported through the BBB was seven times higher in the transferrin–folate doxorubicin-loaded liposome category of glioma bearing rates than in the non-targeted doxorubicin-loaded liposome group of glioma bearing rates [44]. Multiple biological targets have been published in the treatment of glioblastoma, including hyaluronic-acid-conjugated liposomes, c(RGDyK)-pHA-PEG-DSPE-incorporated DOX-loaded liposomes, curcumin-loaded RDP–liposomes, iron oxide nanoparticles coated with CTX loading plasmid DNA ciphering TRAIL, and chitosan–PEG–polyethyleneimine copolymers. These compounds are successful against glioblastoma. On the other hand, the in vivo findings revealed a possible use in diagnosis with substantial therapeutic consequences, leading to a remarkable accumulation in the brain tumor regions [77].

According to Arial et al., DOX-loaded liposomes (liposome–DOX) and DOX-loaded microspheres (microsphere–DOX) were prepared and combined with TGP for prolonged drug delivery, and their impact on cell feasibility was investigated using glioma cell lines (LN229 and U-87MG). DOX-loaded TGP (TGP–DOX) was used in combination with microsphere–DOX (microsphere–DOX + TGP) or TGP combined with liposome–DOX (liposome–DOX + TGP) for in vitro studies in LN229 + U-87MG cells. TGP–DOX lowered cell survival by 29% and 91% on days 10 and 1 of treatment, respectively. TGP had no toxic impact on LN229 or U-87MG cells alone, suggesting that DOX mediated its anti-tumor consequence after being released from the TGP gel. In addition, DOX release was prolonged (10–30 days) in TGP + liposome–DOX and TGP + microsphere–DOX [77]. In the clinical translation of nanoliposome, the addition of stabilizers and surfactants in nanoliposome synthesis has been offered as a good option to promote electrostatic repulsion, which minimizes the loss of encapsulated drugs and increases the size of the vesicles [78].

The anticancer drugs loaded in the nanoliposomes significantly passed through the BBB in large amounts when compared to the unloaded drugs, suggesting that nanoliposomes are potential nanomedicines for brain cancer targeting. The in vivo investigation showed that nanoliposomes are useful for diagnosis with interesting therapeutic outcomes due to their ability to accumulate in the brain cancer sites. The in vitro drug release experiments demonstrated that anticancer drugs can be released from the nanoliposomes in a slow and sustained manner for a prolonged time, and this mechanism can result in overcoming multidrug resistance and drug toxicity to the normal cells.

### 5.3. Polymer–Drug Conjugates

Polymer–drug conjugates, also termed “polymer prodrugs”, are nanocarriers that comprise four components: polymer backbone, bioactive agent, targeting moiety, and solubilizing agent. The model of polymer–drug conjugates was firstly proposed in 1975 by Helmut Ringsdorf (Figure 5) [79]. The solubilizing agent and targeting moiety are incorporated into the polymer prodrugs to enhance the therapeutic and pharmacokinetic outcomes of the loaded drugs. The advantages of polymer prodrugs include reduced drug toxicity, improved drug solubility, enhanced bioavailability and biodegradability, excellent biocompatibility, and improved pharmacological properties. Furthermore, they possess the ability to preserve and protect loaded bioactive agents from enzymatic attacks during circulation and they deliver the therapeutic agent to the target biological site [80].

Carbon nanotubes with polyethene glycol-linked conjugates of mangiferin were synthesized. On U-87 cell lines, cytotoxicity investigations and flow cytometry were performed. At the pH of cancer cells, drug release experiments showcased a spatiotemporal release pattern. The IC50 value was reduced by 1.28 times in cytotoxicity studies, suggesting successful anticancer activity, while the hemolytic profile demonstrated safety. In contrast to the free drug, the flow cytometry indicated that the nano-conjugate effectively induces apoptosis with minimal necrosis. The pharmacokinetic study confirmed that the increase was multiplied by four in the region under the curve, i.e., the bioavailability of the drug afterwards conjugation to that of plain mangiferin. Based on the findings, it was hypothesized that these functionalized nanocarriers can deliver phytochemicals to brain cancer cells efficiently and safely [81]. There are few reports on the clinical translation of polymer therapeutics. Successful clinical translation of polymer therapeutics will require rigorous optimization and a high throughput assessment of the designed conjugates. Measurable design specifications are also essential [82].

One challenge in the clinical translation of polymer–drug conjugates is the uncontrolled conjugation of drug molecules into the polymeric carriers, resulting in heterogeneity, varied drug loadings, etc. However, new controlled synthetic approaches have been developed for quantitative drug loading efficiencies. In the conjugation of two or more drugs for combination therapy, maintaining an optimal ratio of drugs is challenging in vivo, due to the different pharmacokinetics and biodistribution of the drugs [83].

### 5.4. Solid Lipid Nanoparticles

Solid lipid nanoparticles (SLNs) are hydrophilic nanoparticles whereby the bioactive agents are encapsulated in their core (Figure 6) leading to slow drug release kinetics. They also exhibit reduced toxicity and a large surface area [78]. To deliver the anticancer drug, adriamycin, to tumor tissue, Hodoshima et al. developed lipid nanoparticles. For easy encapsulation in SLNs, easy and economical PEG lipid derivatives (to reduce RES uptake) were synthesized [84]. By acylation, 5-fluorouracil was translated to stearyl prodrug, which improved the drug’s lipophilic properties and, hence, its incorporation into SLNs. Physical agglomeration was used to make SLNs, and the concentration of 5-fluorouracil in mice liver was doubled in the case of SLNs compared to 5-fluorouracil injection [85,86]. DOX–SLNs were prepared using a solvent evaporation procedure with an average particle size of close to 200 nm and a polydispersity index (PDI) of 0.3. On U87MG cell lines, a cytotoxicity study demonstrated that DOX–SLNs were more toxic than plain DOX. The internalization of DOX–SLNs within brain cancer cells was also supported by cellular uptake studies. The design of SLNs as carriers for brain cancer treatment with improved potential to cross the BBB was reported in the study [84,87].

Coating and conjugating SLNs, on the other hand, will improve the amount of drugs delivered to the brain. Curcumin delivery to the brain is efficient when SLNs are coated with PS80 [88]. PS80 tends to allow ApoE adsorption on SLNs in the identical way that PLGA and PBCA nanoparticles do [89]. The surface functionalization of SLNs has also been widely applied to optimize SLN transmission to the brain. In contrast to the molecules alone, SLNs functionalized with CBSA [90], transferrin [91], angiopep-2, thiamine, and ApoE demonstrated higher delivery of their cargos to the brain. Dal Magro et al. investigated the brain bioavailability of SLNs functionalized with an ApoE-derived peptide in mice based on the nanoparticle administration path. The nanoparticles were delivered either intraperitoneally, intravenously, or intratracheally [92]. After intravenous and intratracheal administration, the SLNs could be seen in the brain using in vivo fluorescence molecular tomography, revealing their capability to cross the BBB. The clinical translational requirements include a size range of 100 nm or less, adequate product stability, robustness, extremely effective encapsulation techniques, scalable manufacturing procedures, and a minimum surface charge [93,94].

The anticancer drugs loaded in SNLs showed high cytotoxicity against brain cancer cells when compared to plain drugs, indicating that SNLs can significantly improve the anticancer activity of selected anticancer agents. SNLs can further improve the antitumor efficacy of anticancer drugs by increasing cellular internalization within the brain cancer cells. These nanomedicines can cross the BBB, demonstrating that they can be very useful in the treatment of glioblastoma and other related brain conditions. Furthermore, the high drug loading and encapsulation efficiency of SNLs can lead to a high amount of drugs delivered to the brain.

### 5.5. Nanostructured Lipid Carriers (NLCs)

Nanostructure lipid carriers (NLCs) belong to the group of lipid nanoparticles together with SLNs. They comprise a combination of solid lipids and spatially incompatible liquid lipids (Figure 7) and they have the advantage of enhancing drug loading ability and bestowing release properties [95]. NLCs are usually formulated using oil, whereas SLNs are formulated using organic solvents. Recently, another group concentrated on developing NLCs as drug delivery carriers for the treatment of glioma. For example, NLCs modified with the arginine–glycine–aspartic acid peptide (RGD) were applied to deliver TMZ for the treatment of gliomatosis cerebri [96]. The in vivo studies of DNA and TMZ, i.e., co-encapsulated NLCs for GBM, was examined using mice as an animal model. RGD-TMZ-NLCs were found to have a higher cytotoxic effect on U87MG cells than TMZ-NLCs, as well as a greater antitumor efficacy in vivo. The results indicated that DNA-NLCs/TMZ improved anti-tumor potential and gene transfection efficacy by transferring the two of the drugs and gene into the gliomatosis cerebri [97]. To tackle this issue, Kim et al. [98] designed RIPL peptide NLCs conjugated with cleavable PEG that was released from the nanocarrier surface at the tumor position in an acidic environment. The hydrophobic anchor 1,2-dipalmitoyl-sn-glycero-3-phosphothioethanol (DPPE) was used to establish cleavable PEG, which was connected to the PEG3000 chain through a hydrazone bond [99].

Tsai et al. used an intravenous route to evaluate the brain targeting capacity of baicalein-loaded NLCs loaded with poloxamer and vitamin E [100]. According to the researchers, vitamin E was found to help increase the stability of baicalein in vivo. Hsu et al. reported that by changing the composition of NLCs, they were able to effectively target the brain. In vivo bioluminescence monitoring revealed that intravenously injected combined polyethylene glycol NLCs and polysorbate 80 significantly enhanced brain targeting when compared to an aqueous apomorphine solution [101]. In contrast, curcumin NLCs exhibited improved cytotoxicity in the astrocytoma–glioblastoma cancer cell line. In vivo biodistribution findings indicate that intranasal administration of curcumin NLCSs resulted in a greater concentration of drug in the brain than drug solution. According to the study, NLCs appear to be a good drug delivery system for the treatment of brain cancer [23].

The poly (d,l-lactic-co-glycolic acid) nanoparticles loaded with bevacizumab were designed and administered intranasally to CD-1 mice to study their pharmacokinetic and pharmacodynamic profiles, after almost 7 days of administration. In comparison to free bevacizumab administered intranasally, the PLGA NP demonstrated significantly greater brain bioavailability. PLGA NPs loaded with bevacizumab improved bevacizumab penetration (higher Cmx) and residence time in the brain. The efficacy of this nanosystem was studied in an orthotopic GBM nude mice model, testing tumor growth and anti-angiogenic impact using bioluminescence. After 14 days, PLGA NPs loaded into bevacizumab exhibited a remarkable reduction in tumor growth as well as a stronger anti-angiogenic effect. These findings can be supported by the fact that, after 14 days of formulation administration, only the PLGA NPs loaded into the bevacizumab group promoted the uptake of bevacizumab into the brain in vivo [102].

The experimental data have demonstrated that NLCs can be explored in combination therapy for the treatment of brain cancer. The NLCs loaded with anticancer drugs show high cytotoxicity against glioma cells, as well as an improved antitumor activity than plain anticancer agents, suggesting that NLCs are potential nanomedicines for brain cancer therapy. The polymeric NLCs significantly improved brain targeting than the free anticancer drugs, indicating their capability for brain cancer targeting. The in vivo studies exhibited that NLCs result in greater bioavailability and high accumulation of the drug in the brain when compared to pristine drug solutions. In addition, NLCs loaded with anticancer drugs demonstrate a remarkable reduction in tumor growth, revealing that the NLCs can be effective candidates to overcome the emergence of multi-drug resistance, which is common in the currently used anticancer drugs. The findings are likely to provide an unconventional path to a more secure and efficient delivery system. Methods for scaling up their production, as well as their use in clinical trials in the coming years, should be clinically examined. The findings are likely to provide an unconventional path to a more secure and efficient delivery system.

### 5.6. Thermosensitive Gel

Gels are polymeric nanomedicines that possess three-dimensional networks (Figure 8) and they are prepared from natural and synthetic polymers [103]. They can absorb and preserve huge volumes of water and biological fluids. The porosity of polymer-based gels is influenced by features such as formulation procedure, polymer composition, etc. The advantages of gels include non-toxicity, non-immunogenicity, excellent biocompatibility, and affordability [103]. Ding et al. designed calcium phosphate nanoparticles made of polyethylene glycol-dipalmitoyl phosphatidylethanolamine as an injectable thermo-responsive hydrogel for local as well as sustainable delivery of TMZ and PTX to glial tumors. TMZ and PTX were loaded into nanoparticles using the double emulsion process. The anti-glioma impact of TMZ/PTX was (1:100) on C6 cells. The thermo-responsive gel suppressed glioma growth by autophagy, while PTX and PTX-TMZ nanoparticles hindered glioma cell proliferation [104].

To deliver ellagic acid (EA) for the treatment of brain cancer, Kim et al. prepared thermosensitive gel glycerophosphate chitosan loaded with EA. The Ch/-GP solution generated a heat-induced gel at body temperature. In the existence of lysozymes, the in vitro release rate of ellagic acid from an EA-loaded Ch/-GP gel was 2.5 times greater than in the exclusion of lysozymes. Besides that, the anti-tumor activity of EA-loaded Ch/-GP gel was examined using C6 rat glioma cells and human U-87 glioblastoma cells. Three days after incubation, the cell feasibility of C6 rat glioma and U-87 cells was lower than that of the chitosan gel. The efficacy of EA-loaded dialyzed chitosan solution gel on human rat C6 glioma and U-87 glioblastoma cells in an EA concentration-dependent manner was studied. When the concentration of EA in DCh/-GP gels was increased, the metabolic activities of both cells were reduced [105,106]. Unexpected leakage at close physiological temperature might have unfavourable consequences, including systemic toxicity and off-target tissue interactions. To avoid these dangers, it is critical to optimize the formulation, which necessitates the employment of a model drug to precisely define the formulation [107].

The polymeric thermosensitive gels display the ability to be used in combination therapy, which is one of the best strategies to overcome the limitations of presently used anticancer drugs, especially drug resistance. The studies have demonstrated that the drug-loaded gels can significantly suppress glioma growth, while combination therapy using gels can inhibit glioma cell proliferation, suggesting that thermosensitive gels are potential nanotherapeutic for the treatment of brain cancer. The in vitro drug release profiles displayed that the bioactive agents are released from the thermosensitive gels in a controlled and sustained manner. Furthermore, the polymeric thermosensitive gels significantly improved the antitumor activity of loaded anticancer agents on brain cancer cell lines when compared to free agents, demonstrating that they are effective candidates for the treatment of cancer.

### 5.7. Dendrimers

Dendrimers are polymer-based nanomedicines that are monodispersed, high-branched, and possess three-dimensional structures (Figure 9) [108]. Dendrimers are very important in drug delivery because of their low polydispersity, controlled molecular weight, and good biocompatibility. The functional groups on the outside layer of polymer-based dendrimers can be used for the incorporation of bioactive agents and targeting moiety. Their intramolecular cavity is very useful for encapsulating drugs, leading to sustained drug release profile, improved drug biological activity, and reduced drug toxicity [108]. Peripheral functional groups and internal cavities in PAMAM dendrimers are the most thoroughly studied dendrimers and can be adjusted to encapsulate agents or other cargoes for biomedical uses. Dendrimers may be used as nano-agents to treat tumors, bacteria, and viruses infections [109]. The cytotoxicity of PAMAM dendrimers terminated with positively charged groups is influenced by the generation and concentration in vitro. PAMAM dendrimers of the G5 or lower generation are non-toxic [110]. Sarin et al. prepared doxorubicin conjugated into gadolinium-chelated G5 PAMAM dendrimer via pH-sensitive covalent correlations (Gd-D5-DOX) as a theranostic for brain tumors. Gd-D5-DOX with a diameter of 7–10 nm was shown to deliver therapeutic DOX concentrations across the BBB into brain tumor cells [111]. The findings concluded that one dose of Gd-G5-DOX was way more successful in preventing the growth of RG-2 glioma than free DOX at an equal dose in preventing the growth of RG-2 glioma [111].

Zhao et al. developed fibrin-binding peptide CREKA conjugated to PAMAM dendrimers to create a small nanoparticle DDS that targets extracellular fibrin in brain tumors to enhance nanoparticle retention. The CREKA-modified PAMAM accumulated more and penetrated more in glioblastoma multiforme tissue than unmodified CREKA PAMAM [112], indicating that it may be a promising treatment option for brain tumors. Besides that, other brain tumor capillaries overexpress multiple receptors, which guide ligand-anchored dendrimer-based DDSs and promote drug delivery to the brain tumor tissue [113]. DOX with PEGylated PAMAM dendrimer was conjugated to PAMAM through the acid-sensitive cis-aconityl linker [114], which can enhance tumor targeting by linking to overexpressed integrin receptors on the tumor cells and regulating doxorubicin release in acidulous lysosomes [115]. Hydroxyl-modified generation-four (G4-OH) PAMAM dendrimers administered systemically were taken up significantly in vivo in the activated microglia and astrocytes of the brain of newborn rabbits with cerebral palsy. The uptake was not significant in the healthy rabbits. *N*-acetyl-L-cysteine-based dendrimers suppressed neuroinflammation in brain injury with improved motor function [116]. The degree of glial activation, BBB damage, and severity of brain disease all influence dendrimer accumulation in the disease site [117]. The breakdown of occludin, TJ proteins zonula occludins, and claudin-5 in the periventricular region due to neuroinflammation induced the uptake of dendrimers into the brain parenchyma. However, there is an increased possibility of transcytosis due to inflammation [118].

The toxicity of Aβ was substantially decreased by 27 terminal morpholine groups which beautifully decorated the gallic acid triethylene glycol dendrimer, most likely by accelerating the procedure of fibril formation and reducing the quantity of prefibrillar forms in the system [119]. Phosphorus dendrimers and PAMAM can modify amyloid formation. Generation-five (PPI-G5-Mal) and generation-four (PPI-G4-Mal) maltose-decorated PPI dendrimers were nontoxic to SH-5YSY and PC12 neuroblastoma cells and could interact with Aβ fibrilization [120]. The architecture and size of dendritic nanoparticles have an impact on their capability to deliver drugs to the brain and should be considered when designing brain-targeted drug delivery systems. Dhanikula et al. developed a series of polyether–copolyester dendrimers with a series of branching structures [121]. Based on an in vitro model, the study found that the architecture of dendrimers impacts the cellular update of dendrimers and their permeability through the BBB [121]. The internalization of the synthetic PEPE dendrimers was reported to be primarily mediated by clathrin and caveolin-mediated endocytosis. Moreover, the PEPE dendrimers were able to pass through the BBB in large quantities in vitro without disturbing the close junctions [121]. Their findings indicate that this new form of dendrimer can be further tested for drug delivery in the brain. Another major obstacle in getting drugs to brain tumors is the blood–brain tumor barrier. Dendrimer nanoparticles with a diameter of less than 11.7 to 11.9 nm have been reported to be able to pass through the vents of the blood–brain tumor barrier in RG-2 malignant gliomas [122].

PEG was conjugated to the PAMAM dendrimer. The resultant copolymer (PEG-PAMAM) was designed to deliver DOX and DNA [118]. Benzyl ester dendrimer silicon phthalocyanine was prepared to make polymeric nanoparticles and encapsulated in amphiphilic block copolymers. Confocal laser scanning imaging was used to analyze cellular absorption and subcellular localization in U251 glioma cells. After being treated for 6 h, the cellular uptake of U251 glioma cells reached its peak and was found in lysosomes and mitochondria [123]. The effectiveness of photodynamic therapy against U251 glioma cells was tested. Under laser irradiation, the analysis indicated great photo-cytotoxicity in U251 glioma cells. This polymeric nanoparticle has the potential to be a good candidate for glioma therapy [124]. Due to concerns about toxicity and biocompatibility, the clinical translation of dendrimer-based drug delivery devices has been restricted. Dendrimers have indeed demonstrated a strong affinity for lipids, metal ions, bile salts, and nucleic acid proteins, causing toxicity. As a response, great effort is being put into designing biocompatible dendrimers with surface modifications to improve their biocompatibility. The complexity and cost involved with dendrimer manufacture must be addressed [124].

The particle size analysis demonstrates that polymeric dendrimers possess suitable particle size to pass through BBB by either transcellular or paracellular pathways to deliver loaded bioactive agents to the brain tumor without disturbing the close junctions. The drug-loaded dendrimers display high inhibition of glioma cell growth and proliferation when compared to free drugs, indicating that the dendrimers can result in improved therapeutic outcomes of the anticancer drugs on brain cancer cells. Dendrimers are potential nanocarriers to promote drug delivery to the brain tumor environment. The in vitro experiments of drug-loaded dendrimers exhibit higher cellular uptake and internalization by brain cancer cell lines with improved cytotoxicity when compared to plain anticancer drugs, confirming dendrimers as effective nanomedicines for glioblastoma therapy.

### 5.8. Micelles

Micelles colloidal and self-assembled nanoparticles (Figure 10) with an average particle size that range between 5 and 100 nm [125]. They are made up of amphiphiles or surfactants and consist of two different regions: hydrophilic head and hydrophobic tails. The concentration that results in the production of micelles is called critical micelle concentration (CMC) [126]. Various factors affect the formulation of the micelles, such as the size of the hydrophobic domain in the amphiphilic molecules, the concentration of amphiphiles, the used solvent, and temperature. The advantages of micelles in biomedical applications include high drug cellular uptake, easy elimination from the biological environment after biodegradation, high drug encapsulation, and high drug loading capacity, enhanced drug stability. Furthermore, they can be used in combination therapy, can protect normal body cells from drug toxicity, and can enhance drug pharmacokinetic parameters [127]. Nanoparticles functionalized and GMT8 aptamer greatly improved tumor spheroid infiltration and intracellular drug delivery in U87 tumor spheroid uptake and in vitro cell uptake research. The in vivo imaging analysis indicate that ApNP could target glioblastoma in an orthotropic brain glioblastoma model, leading to a two-fold greater tumor uptake than non-targeted controls in an orthotropic brain glioblastoma model [128].

Soni et al. generated nanogels by crosslinking polymeric micelles made of N-isopropylacrylamide and N-vinylpyrrolidone for the delivery of N-hexylcarbamoyl-5-fluorouracil which is poorly water-soluble [129]. They found that coating the drug-carrying nanogels with polysorbate 80 increased N-hexylcarbamoyl-5-fluorouracil aggregation in the brain [129]. Even though a substantial portion of N-hexylcarbamoyl-5-fluorouracil-loaded nanogels accumulates in the RES coating nanogels with polysorbate 80, the quantity of nanogel uptake in the brain increased from 0.18 per cent to 0.52 per cent of the injected dose. Thus, the covalently coupling of drugs to the micellar nanoparticles has been examined for brain drug delivery. Aspartic acid was conjugated to doxorubicin (DOX) trace of a poly (ethylene glycol)-β-poly (aspartic acid) block copolymer by Inoue et al. [130]. CED was used to deliver the polymeric micelles generated by the DOX-conjugated poly (aspartic acid) block to the brain. In comparison to liposomal DOX and free DOX at the same dosage (0.2 mg/mL), micellar DOX injected by CED resulted in a longer median survival (36 days) [130]. MPEG-PCL block copolymers were prepared by ring-opening polymerization of -CL in the existence of MPEG. The ninhydrin reaction, which turns reddish-violet when it reacts with the amino residue, was used to confirm the Tat analogue conjugation with MPEG-PCL through the ester bond. MPEG-PCL was 50 nm with a small negative charge in water when it was not loaded. Non-loaded MPEG-PCL-Tat, on the other hand, was around 30 nm with a positive charge. Moreover, coumarin-loaded MPEG-PCL has a loading capacity of 70% and is about 100 nm in size, even though MPEG-PCL-Tat loaded into coumarin is about 90 nm and has a drug loading efficacy of 90% [131]. They are useful platforms for loading hydrophobic drugs. The clinical translation of polymeric micelles has been possible for the administration of drugs via the parenteral routes [132]. There is a critical need to develop mucoadhesive micelle formulations for the administration of drugs via the mucosal route.

The in vitro experiments show that polymer-based micelles can improve the cellular uptake of the loaded anticancer agents within the brain cancer cell lines. Other studies have demonstrated that drug-loaded micelles possess enhanced drug targeting to the brain tumor than the free anticancer drugs.

## 6. The Limitations of Nanomedicine

Nanomedicine, despite its numerous advantages and numerous applications, it cannot be defined as being ideal. There have been reports of toxicity associated with nanomedicines, such as blood clotting, as well as hemolysis caused by cationic nanoparticles and brain damage inducing lipid peroxidation by fullerenes [133]. In vitro studies on the newly engineered nanotubes also revealed ROS formation, changes in cell morphology, oxidative stress, mitochondrial dysfunction, and lipid peroxidation, ultimately leading to toxicity. Carbon nanotubes also clumped the airways hindering the lungs from accessing sufficient oxygen. Furthermore, the non-biodegradable nanocarriers can cause soil, water, and air pollution [134]. Another limitation associated with nanomedicines is their high cost. The application of nanomedicine can result in a high cost of health care, making it unaffordable to the poor [135].

Nanoparticles rely on the surrounding that drives their uptake resulting in size changes and toxicity. Solubility, surface charge, chemical composition, and the modification of functional groups on the nanoparticles are all factors that can contribute to their toxicity [136]. Unknown interactions of these nanocarriers inside the body can result in unpredictable reactions, entering into capillaries, to other body parts, causing toxicity. They can also access various cell organelles causing damage [134]. The biodegradable nanoparticles are easily excreted, but the non-biodegradable ones can accumulate in organs, causing harm and inflammation resulting from a build-up at the site of drug administration [137].

## 7. Strategies under Development to Overcome Limitations of Nanomedicine

In cancer, drug resistance refers to the potential of cancerous cells to become tolerant to therapies that would have killed them previously. Multiple factors contribute to the development of drug resistance; some are based on evolution and spontaneous mutations (intrinsic), while others are a result of drug uptake (acquired or extrinsic) [138]. Nanocarriers containing anticancer drugs were found to enter the cell through endocytosis, leading to drug release at a perinuclear location within the cell, far from cell membranes and efflux pumps. Because of the overall assessment, the loaded drugs bypass efflux transporters [139,140]. Combination therapy, in which many medications are loaded into a single drug carrier, is another way to treat drug-resistant tumors with NPs. Developing nanocarriers can be used to incorporate efflux pump inhibitors and chemotherapeutics can be used to impede the function of efflux transporters [141,142]. Hypoxia [143], which results from increased oxygen demand by proliferating cancer cells caused by abnormal blood vessels, is another factor that contributes to multidrug resistance. This factor causes tumor heterogeneity and induces a more aggressive phenotype which can mediate the overexpression of drug efflux proteins [144]. HIF-1α overexpression has been described in a variety of human cancers [145], and targeting HIF-1α is yet another treatment way to overcome drug resistance [146]. Drug resistance in cancer was successfully overcome by using nanocarriers loaded with HIF-1siRNA [147,148].

Immune cells can extravasate after the BBB is disrupted with microbubble-mediated focused ultrasound (FUS). In vivo studies are now investigating the possibilities of immunotherapy in combination with FUS [149]. The noninvasive nature of FUS in combination with numerous drugs makes FUS a versatile and promising technique for drug or immune therapy delivery for various brain tumors. The non-specificity of drugs can lead to toxicity and is a possible concern with intranasal drug delivery. Toxicity can be reduced by targeting tumor cells. GRN163 is an example that has been studied in vivo and has been shown to specifically target telomerase. The treatment resulted in precise tumor targeting as well as less significant side effects [150]. Combining intranasal drug delivery plus microbubble-mediated FUS is another technique to reduce toxicity to surrounding brain tissue. The combination of these approaches resulted in higher and targeted drug uptake in the tumor location [151]. Because only a few studies have looked into the use of intranasal drug delivery for the treatment of primary brain tumors, it is difficult to say whether intranasal drug delivery is a good way to get around the BBB. Drug modification may be performed by chemically enhancing a pharmaceutically active drug to create a pharmacological active prodrug that can improve the BBB permeation. Another strategy is the lipophilic character of the medication that allows the drug it to diffuse well through the BBB [152]. While making a more lipophilic prodrug or increasing the lipophilicity of a drug molecule can help it cross the BBB, it can also increase its metabolism and active clearance by efflux proteins, which are both disadvantages of this technique [153].

There are some shortcomings associated with focused ultrasound (FUS). One major shortcoming of ultrasound is it can be significantly reduced by the bone and its application in the brain can distort ultrasound waves by the skull [154,155]. It also gives room to the uptake of unwanted substances, such as inflammatory agents and foreign matter into the brain. Due to the aforementioned limitation, there is a pressing need for clinicians to fully study and understand tissue responses to FUS, which includes localized edema, hemorrhage, etc., [156]. FUS is a less invasive technology useful for making a thermal lesion in the brain without making a hole in the skull. In the past, the scattering and refraction properties of ultrasound limited its use. Currently, the use of sophisticated computer algorithms makes it useful to produce a small lesion in the needed part of the brain [157]. FUS has been used to treat Parkinson’s disease and various forms of tremor. However, it is not appropriate in the treatment of movement disorders where deep brain simulation is used [158]. FUS is cost-effective with no risk of infection, skin incision, or implants, though it does make a lesion with irreversible effect [158]. The use of MR-guided FUS is limited when used for the treatment of tumors in moving body organs due to its sensitivity to motion artifacts, such as in the treatment of liver tumors. To control movement such as respiratory movement, general anesthesia is employed for intermittent breath holding [159]. Organ movement during the use of a high-intensity focused ultrasound surgery can result in an incomplete target ablation or collateral damage [160].

## 8. Future Perspective and Conclusions

The most widely studied nanoparticles are spherical; nevertheless, nanoparticles can be created in a variety of shapes, including cubes, rods, discs, and stars [149]. The shape of nanoparticles affects their behaviour in blood flow, endothelial cell interactions, and the MPS, impacting cellular uptake, circulation time, and biodistribution [149].

Liposomes’ distinctive phospholipid bilayer structure, which is comparable to that of a natural membrane, has made it easier for medicinal compounds to pass across the BBB and reach the brain [150]. Because of their extensive track record, minimal toxicity, and capacity to transmit both hydrophilic and lipophilic chemicals, they are the most studied and clinically recognized nanocarriers [150]. Hu et al. revealed glutathione PEGylated liposomal doxorubicin (2B3101), a brain-targeted liposomal formulation that has reached and been tested in clinical trials [151]. Glutathione was used as a BBB-targeting ligand. In patients with gliomas, 2B3101 has completed a Phase I/IIa trial [151].

Soft lithography technologies can also be used to create non-spherical nanoparticles (such as cylindrical shapes) [152]. Surface adjustments can be applied through the terminal carboxylic acid groups, for example, to create diblock (PEG-b-PLGA) or triblock (PLGA-b-PEG-b-PLGA) [153] copolymers, or to introduce targeted moieties, such as folic acid or antibodies [161]. For bridging the BBB, a variety of PLGA formulations has been investigated [162]. For example, in transgenic mice, PLGA nanoparticles coated with a cyclictransferrin-targeting peptide and loaded with A*β* generation inhibitor peptide and curcumin exhibited improved spatial memory and recognition [163]. In addition, two non-CNS targeting PLGA formulations have been authorized in clinical trials. Genexol-PM was authorized in South Korea in 2006 for the treatment of head, neck cancer, and breast cancer, while Nanoxel was approved in India in 2007 for a variety of malignancies [149]. Drug delivery for neurological illnesses has also been explored using PCL-based nanoparticles [164]. For instance, in an in vitro BBB model, peptide-functionalized PEG-PCL micelles showed considerably higher transport ratios and greater accumulation in an intracranial glioma tumor-bearing in vivo model [165,166].

The anti-transferrin receptor antibody was conjugated to the polymeric nanoparticles and rod-shaped nanoparticles accumulated seven times more in the brain than spherical ones [167]. In a microfluidic model, rod-shaped polystyrene nanoparticles had a lower affinity than spherical nanoparticles, but they had a higher BBB transit when normalized by endothelial attachment [167]. Findings with a similar shape were also noted for inorganic hybrid nanoparticles with a polymer shell. For instance, gold nanorods coated with PEG and RVG (a BBB shuttle peptide) accumulated more in the brain of mice than spherical nanoparticles [167]. Some shapes, including cubic nanocages, nanostars, and nanodiscs, have yet to be studied for BBB penetration efficacy, although they have exhibited shape-dependent biodistribution and cell attachment, which could contribute to increased BBB penetration. For instance, in a tumor-bearing animal model, gold nanoparticles with a nanodisc shape were reported to have better tumor penetration than nanorods and nanocubes [168,169].

However various concerns should be addressed in the future to reach a successful clinical translation of nanoformulations: (a) an eco-friendly green method should be designed for the preparation of nanoformulations; (b) the nanomaterials should be biodegradable in nature and give effective and very safe brain-targeted drug delivery systems; (c) a non-invasive alternative approach for nanocarrier drug delivery should be designed to avoid complications, such as poor patient compliance associated with i.v. and other invasive pathways, and newer drug administration pathways for nanocarrier-mediated CNS drug-delivery systems, such as transbuccal, oral, nasal, or mucosal/sublingual, need to be investigated; and (d) factors, such as shape, size, charge, and moiety, attached to nanomaterials should be well elucidated and studied, which is essential for bypassing and designing brain-targeted drug delivery systems [170].

Nanomedicines are promising therapeutics for the treatment of brain cancer. Their unique features in the biological environment and their size range promote their enhanced cellular uptake. The ability of nanomedicines to improve the efficacy of existing clinically authorized anticancer medications by boosting the efficiency or specificity of delivery to their sites of action are their most significant advantage. Moreover, it is easy to add biological targeting moieties to the surface of nanomaterials, thereby enabling targeted delivery of chemotherapeutics to malignant tissue while lowering toxicity in healthy tissue. Studies have also shown that changing the size and shape of particles, as well as procedures to modify their surface, can result in the formation of nanoparticles with the desired properties, but without a toxic effect. In vitro studies showed that specific oxide nanoparticles can act selectively on cancer cells with lower toxicity than normal cells. Polymer coatings are an effective way to reduce the toxicity of nanoparticles. To drive the clinical translation, more information around nano delivery to the brain is required, i.e., how a nanoformulation should be properly developed and optimized. Another problem limiting the clinical applicability of brain-directed NCs is the lack of in vivo evaluations in general. Moreover, the distribution of nanomaterials in the brain is another issue. To achieve drug targeting in the brain, the following must be considered: (i) a uniform preparation procedure should be designed to make the NPs more homogenous and predictable; (ii) materials should be biodegradable and capable to be eliminated from the brain, which would provide the brain-targeted drug delivery systems with biological safety; (iii) factors that influence in vivo behaviour of NPs should be well elucidated and investigated, which is fundamental for constructing brain-targeted drug delivery systems; and (iv) targeting efficiency is far from satisfactory and considerable enhancement should be made before their clinical application.

## Figures and Tables

**Figure 1 pharmaceutics-14-01048-f001:**
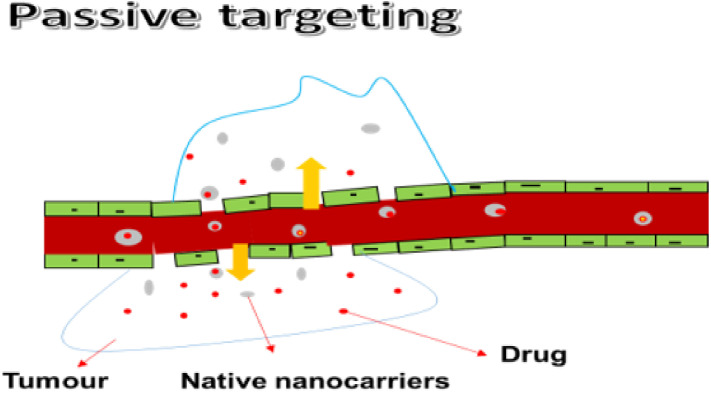
Passive targeting.

**Figure 2 pharmaceutics-14-01048-f002:**
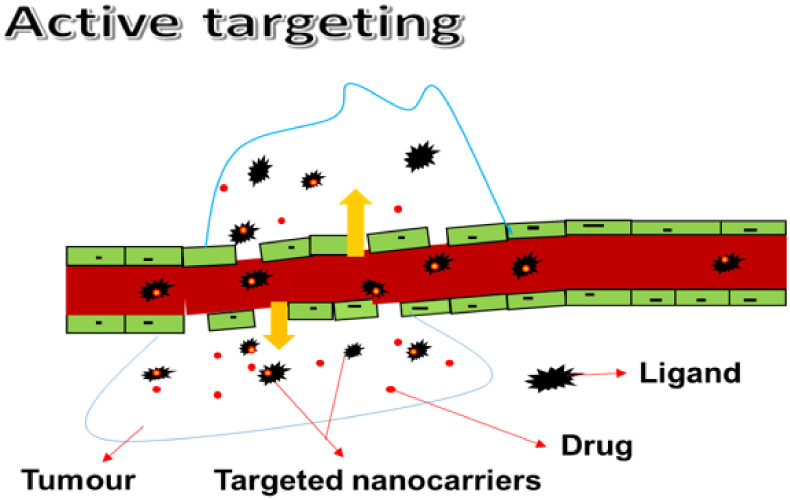
Active targeting.

**Figure 3 pharmaceutics-14-01048-f003:**
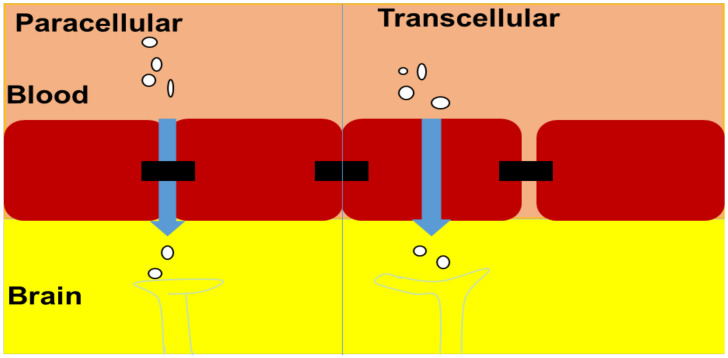
Mode of transportation across the brain.

**Figure 4 pharmaceutics-14-01048-f004:**
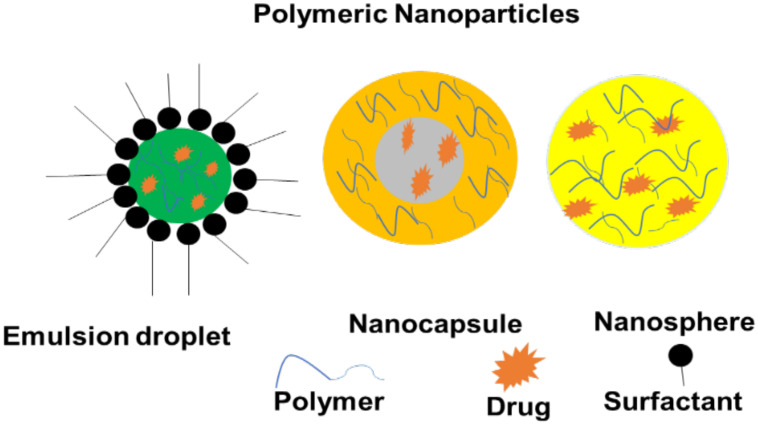
Polymer-based nanoparticles.

**Figure 5 pharmaceutics-14-01048-f005:**
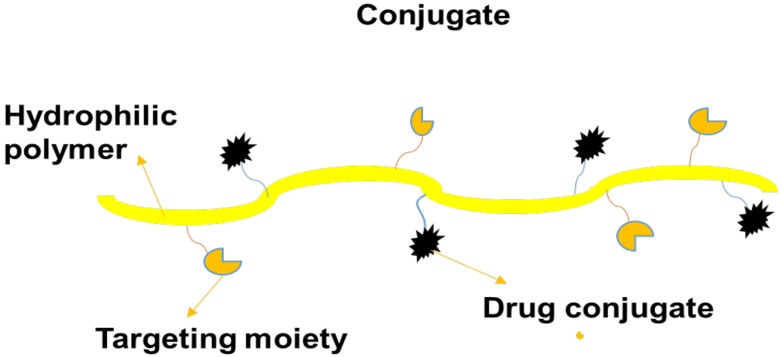
A schematic diagram of a conjugate.

**Figure 6 pharmaceutics-14-01048-f006:**
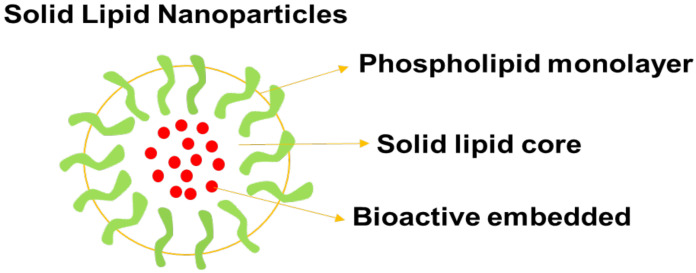
A schematic diagram of solid lipid nanoparticles.

**Figure 7 pharmaceutics-14-01048-f007:**
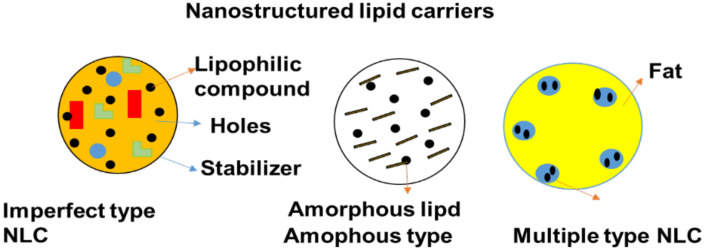
Schematic diagram of nanostructured lipid carriers.

**Figure 8 pharmaceutics-14-01048-f008:**
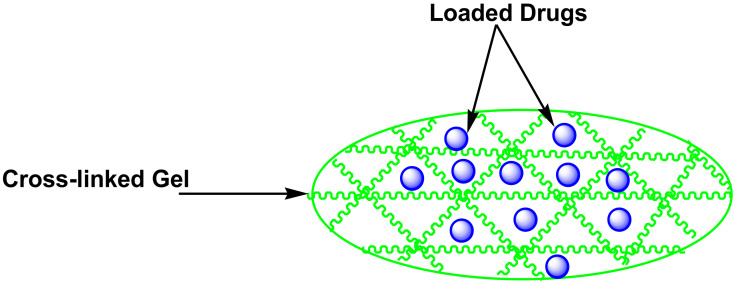
A schematic diagram of thermosensitive gel loaded with drugs.

**Figure 9 pharmaceutics-14-01048-f009:**
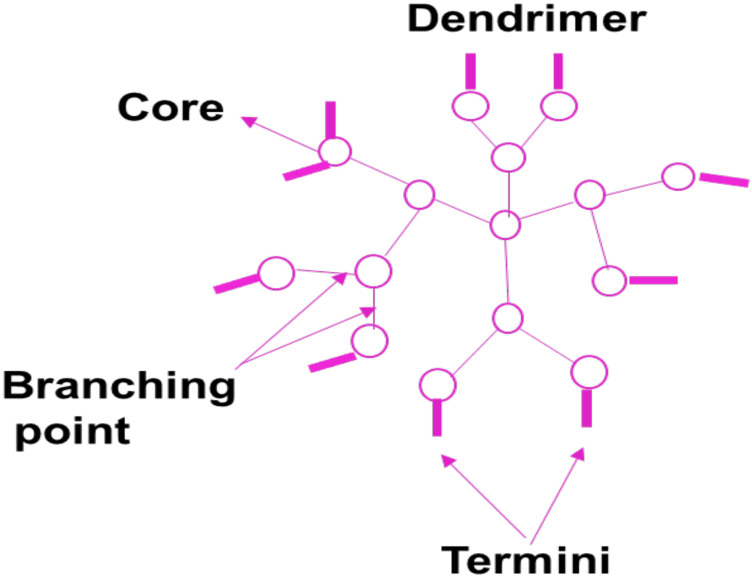
A schematic diagram of a dendrimer.

**Figure 10 pharmaceutics-14-01048-f010:**
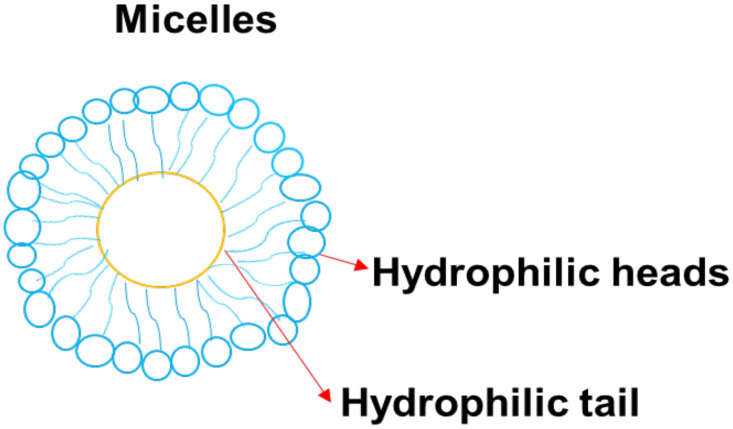
A schematic diagram of micelle.

## Data Availability

Not applicable.

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
