# Peer review of "Efficacy of Polymer-Based Nanomedicine for the Treatment of Brain Cancer"

_pharmaceutics, 2022, doi:10.3390/pharmaceutics14051048_

Round 1
Reviewer 1 Report
Authors have addressed all my concerns and I do not have any more comments.
Author Response
Thank you very much.
Reviewer 2 Report
In the present review article, authors well summarized the different therapeutic targeting strategies, efficacy of nanomedicine and future perspectives in the treatment of brain cancer. Overall, I feel that the present review can attract a lot of attention to develop a novel therapeutic platforms for future brain tumor diagnostics. So I recemond it for publication after addressing the following minor concerns.
- Authors should include recent reaesrch papers especially in 2021/2022.
- Cite the following papers, https://doi.org/10.3390/polym12123055, https://doi.org/10.1007/s11095-021-03122-9, https://doi.org/10.3390/biomedicines8010013
- Authors more concentrated on polymeric based NPs in the present review, so I suggest to authors to rethink about their tittle.
Author Response
In the present review article, authors well summarized the different therapeutic targeting strategies, efficacy of nanomedicine and future perspectives in the treatment of brain cancer. Overall, I feel that the present review can attract a lot of attention to develop a novel therapeutic platforms for future brain tumor diagnostics. So I recommend it for publication after addressing the following minor concerns.
- Authors should include recent reaesrch papers especially in 2021/2022.
- Cite the following papers,https://doi.org/10.3390/polym12123055, https://doi.org/10.1007/s11095-021-03122-9, https://doi.org/10.3390/biomedicines8010013
- Authors more concentrated on polymeric based NPs in the present review, so I suggest to authors to rethink about their tittle.
Author’s response
- Recent papers has been added as reference 5
- Cited as reference 174, 73 and 171.
- We have revised the title of the manuscript to reflect that the manuscript is focused on polymer-based nanomedicines.
Reviewer 3 Report
T.Naki et al. submitted a review manuscript “Efficacy of Nanomedicine for the Treatment of Brain Cancer” and This review reports the in vitro and in vivo results of nano-medicines designed for the treatment of brain cancers. There were about 18,078,957 cases of cancers worldwide according to the 2018 Cancer Registry with approximately 296,851 cases associated with brain cancers. The major challenge in the development of effective therapies for the treatment of brain tumor is the heterogeneity of the brain tumors and the BBB. Nanomedicines have been reported to accumulate selectively in the tumors due to the enhanced permeability and retention (EPR) effects, which takes full advantage of cancer tissue permeable vasculature and reduced lymphatic drainage.
The review showed the mechanism of drug action in the brain. It mainly includes five parts, such as nanodelivery platforms developed for drug delivery,mechanisms of transport through the blood-brain barrier,the different nanomedicines designed for brain cancer therapy,the limitations of nanomedicine, Strategies under development to overcome limitations of nanomedicine and future perspective. The review introduced in detail the research progress and clinical application of nanomedicine in brain cancer in recent years.
Other comments:
- It is suggested that parts "8" and "9" of the review should be discussed together.
- The abstract part should contain the full text, and only the latter part is summarized in the manuscript.
- Please check the format of the paper carefully according to the requirements of the magazine, Is there a space at the beginning of each paragraph?
- In "2.1.4", Add "This pathway is a potential route for the transport of drugs to the brain." at the end of the paragraph
- In "3.1.2", "As a result of external stimuli such as heat, ultrasound, light, and a magnetic field, environmentally sensitive macromolecular drug carriers can release cargo drugs in the targeted tumor tissues." Please briefly introduce the main disadvantages of these technologies so far and the problems to be solved.
- In "7", "The noninvasive nature of FUS in combination with numerous drugs makes FUS a versatile and promising technique for drug or immune therapy delivery for various brain tumors.". Please briefly introduce the disadvantages and challenges of this technology at present.
Author Response
- Naki et al. submitted a review manuscript “Efficacy of Nanomedicine for the Treatment of Brain Cancer” and This review reports the in vitro and in vivo results of nano-medicines designed for the treatment of brain cancers. There were about 18,078,957 cases of cancers worldwide according to the 2018 Cancer Registry with approximately 296,851 cases associated with brain cancers. The major challenge in the development of effective therapies for the treatment of brain tumor is the heterogeneity of the brain tumors and the BBB. Nanomedicines have been reported to accumulate selectively in the tumors due to the enhanced permeability and retention (EPR) effects, which takes full advantage of cancer tissue permeable vasculature and reduced lymphatic drainage.
The review showed the mechanism of drug action in the brain. It mainly includes five parts, such as nanodelivery platforms developed for drug delivery mechanisms of transport through the blood-brain barrier,the different nanomedicines designed for brain cancer therapy,the limitations of nanomedicine, Strategies under development to overcome limitations of nanomedicine and future perspective. The review introduced in detail the research progress and clinical application of nanomedicine in brain cancer in recent years.
Other comments:
- It is suggested that parts "8" and "9" of the review should be discussed together.
- The abstract part should contain the full text, and only the latter part is summarized in the manuscript.
- Please check the format of the paper carefully according to the requirements of the magazine, Is there a space at the beginning of each paragraph?
- In "2.1.4", Add "This pathway is a potential route for the transport of drugs to the brain." at the end of the paragraph
- In "3.1.2", "As a result of external stimuli such as heat, ultrasound, light, and a magnetic field, environmentally sensitive macromolecular drug carriers can release cargo drugs in the targeted tumor tissues." Please briefly introduce the main disadvantages of these technologies so far and the problems to be solved.
- In "7", "The noninvasive nature of FUS in combination with numerous drugs makes FUS a versatile and promising technique for drug or immune therapy delivery for various brain tumors.". Please briefly introduce the disadvantages and challenges of this technology at present.
Author’s response
- We have combined section 8 and 9.
- The abstract is a summary of the review article.
- Thank you for the comments. There is a space at the beginning of each paragraph based on the journal.
- In section 2.1.4, we have added the statement: "This pathway is a potential route for the transport of drugs to the brain’.
- In section 3.1.2 we have included some of the limitations associated with stimuli-responsive drug delivery systems.
- We have discussed the associated with FUS Challenges discussed in line 968-982.